# Active probing to highlight approaching transitions to ictal states in coupled neural mass models

**Vinícius Rezende Carvalho**[1,2,3], **Márcio Flávio Dutra Moraes**[2,4], **Sydney S. Cash**[3], **Eduardo Mazoni Andrade Marçal Mendes**[1,2,4]*

1 Programa de Pós-Graduação em Engenharia Elétrica, Universidade Federal de Minas Gerais, Belo Horizonte, Brazil, 2 Núcleo de Neurociências, Departamento de Fisiologia e Biofísica, Instituto de Ciências Biológicas, Universidade Federal de Minas Gerais, Belo Horizonte, Brazil, 3 Department of Neurology, Massachusetts General Hospital and Harvard Medical School, Boston, Massachusetts, United States of America, 4 Centro de Tecnologia e Pesquisa em Magneto-Ressonância, Escola de Engenharia, Universidade Federal de Minas Gerais, Belo Horizonte, Brazil

* emmendes@cpdee.ufmg.br

## Abstract

The extraction of electrophysiological features that reliably forecast the occurrence of seizures is one of the most challenging goals in epilepsy research. Among possible approaches to tackle this problem is the use of active probing paradigms in which responses to stimuli are used to detect underlying system changes leading up to seizures. This work evaluates the theoretical and mechanistic underpinnings of this strategy using two coupled populations of the well-studied Wendling neural mass model. Different model settings are evaluated, shifting parameters (excitability, slow inhibition, or inter-population coupling gains) from normal towards ictal states while probing stimuli are applied every 2 seconds to the input of either one or both populations. The correlation between the extracted features and the ictogenic parameter shifting indicates if the impending transition to the ictal state may be identified in advance. Results show that not only can the response to the probing stimuli forecast seizures but this is true regardless of the altered ictogenic parameter. That is, similar feature changes are highlighted by probing stimuli responses in advance of the seizure including: increased response variance and lag-1 autocorrelation, decreased skewness, and increased mutual information between the outputs of both model subsets. These changes were mostly restricted to the stimulated population, showing a local effect of this perturbational approach. The transition latencies from normal activity to sustained discharges of spikes were not affected, suggesting that stimuli had no pro-ictal effects. However, stimuli were found to elicit interictal-like spikes just before the transition to the ictal state. Furthermore, the observed feature changes highlighted by probing the neuronal populations may reflect the phenomenon of critical slowing down, where increased recovery times from perturbations may signal the loss of a systems' resilience and are common hallmarks of an impending critical transition. These results provide more evidence that active probing approaches highlight information about underlying system changes involved in ictogenesis and may be able to play a role in assisting seizure forecasting methods which can

**Data Availability Statement:** All Python scripts are available at https://github.com/vrcarva/WNMM_probing.

**Funding:** This work was supported by Coordenação de Aperfeiçoamento de Pessoal de Nível Superior (CAPES - http://www.capes.gov.br/, Finance code 001 to VC), Conselho Nacional de Desenvolvimento Científico e Tecnológico (CNPQ - http://www.cnpq.br - 308129/2017-2 to EM, 307354/2017-2 and 425746/2018-6 to MM), and Pró-Reitoria de Pesquisa - Universidade Federal de Minas Gerais (http://www.ufmg.br/prpq/ - EM). SSC was supported by Tiny Blue Dot Foundation and by NIH-NINDS (K24-NS088568 and R01-S062092) grants. The funders had no role in study design, data collection and analysis, decision to publish, or preparation of the manuscript.

**Competing interests:** The authors have declared that no competing interests exist.

be incorporated into early-warning systems that ultimately enable closing the loop for targeted seizure-controlling interventions.

## Author summary

Epilepsy is characterized by spontaneous and recurrent seizures. Developing a method to forecast the occurrence of these events would ease part of the burden to patients by providing warning systems and identifying critical periods for closed-loop seizure suppressing methods. However, this task is far from trivial and doing this in a clinical setting has been a challenge that is not yet solved. A possible way to achieve this is by using active probing paradigms, in which stimuli are used to highlight otherwise hidden changes in brain dynamics that may lead to seizures. In this work, this approach is tested in a computational model with two coupled neuronal populations. Active probing responses highlighted underlying model parameter changes that led neuronal activity closer to seizures, something that was not detected with the passive observation in most evaluated model settings. Observed feature changes revealed higher excitability, longer recovery times from stimuli, and increased synchrony between populations as the threshold to seizure-like activity approached. This indicates that low frequency stimuli may be used to reveal early-warning signs and assist in seizure forecasting methods.

## Introduction

The identification of robust electrophysiological features indicating approaching transitions from interictal to ictal states is one of the main challenges in epilepsy research [1,2]. The extraction of markers heralding the imminence of a seizure would be an important step for elucidating the mechanisms of epilepsy and for the development of seizure-prediction methods. The latter would alleviate one of the greatest burdens afflicting epilepsy patients, which is the unpredictability of seizures [3]. Achieving this would be a breakthrough to improve patients' quality of life, whether by serving as seizure advisory systems [4] or by closing the loop to enable time-specific interventions to prevent upcoming ictal events [5].

Although several articles have reported promising seizure-prediction methods since the 90s, several of these have been questioned after the impossibility of reproducing the original positive results [6–8]. Such discrepancies occurred primarily in methods that were originally optimized and tested in specific datasets but failed to achieve similar performances when confronted with a greater variety of long-term EEG recordings. Achieving a sufficiently robust and reliable seizure forecasting method has thus been shown to be a task that is far from trivial [9] and has been described as "the long and winding road" [6].

In addition to being robust to the high number of brain states (sleep, attention, cognition, etc.), the development of seizure-prediction methods should consider that ictogenesis mechanisms may be distinct for different brain structures and different pathologies [6,10]. This motivates the use of individualized methods [11–13]; algorithms should be trained or tuned for each patient in order to account for the innumerous sources of inter-subject variability, which include different epilepsy and seizure types. Other challenges to overcome in developing seizure prediction methods are the scarcity of publicly available long-term EEG databases (although this is changing with the increased use of implantable devices) and the difficulty in extracting features that consistently reflect increased seizure susceptibility–in general, there is

a high level of abstraction from signal features in relation to the underlying mechanisms leading up to seizures [6,14]. A possible approach to overcome some of these problems involves the use of active probing or active observation paradigms [7,15]; by applying stimuli and evaluating responses, more information could be obtained about the state of a system, which would otherwise be inaccessible with its passive observation. Such perturbational approaches have been shown to provide diagnostic value for different conditions and have served as valuable tools throughout the history of neurology and neuroscience research [16]. Altered auditory steady-state responses (ASSRs) have been found in patients with schizophrenia [17], bipolar disorder [18], and temporal lobe epilepsy [19]. The complexity of responses to transcranial magnetic stimulation (TMS) can be used to discriminate levels of consciousness [20] and has shown to have significant diagnostic potential in genetic generalized epilepsy [21]. Visual stimulation, through a binocular rivalry paradigm, reveals slower responses in autism [22]. Tracking cortical excitability in patients with epilepsy has been done with TMS [23,24] and electrical stimuli [25]. Abnormal responses to TMS [26], single pulse electrical stimulation (SPES) and/or cortico-cortical evoked potentials (CCEP) have been used to explore cerebral functional connectivity and assist in delineating epileptogenic cortex for resection [27–29]. However, given the wide variety of epilepsy types and seizure models available, the use of similar interventional or perturbational approaches for seizure forecasting is still limited.

The first use of stimuli to "probe" transitions to ictal states was done by Kalitzin et al. [10], with the use of intermittent photic stimulation (IPS) to test the hypothesis that response changes to these stimuli would precede the onset of seizure activity in photosensitive patients. An increase of the relative phase clustering index (rPCI) was found, which would reflect the hyperexcitability of the underlying dynamical system and would indicate the presence of nonlinear dynamics. Auditory stimuli were used to reveal the pathophysiology of mTLE, where altered neural synchronization induced by monaural pure-tone stimulation [30] or by ASSR [19] may provide useful diagnostic information. ASSRs were also used to analyze ictal and interictal state dynamics in an audiogenic seizure model, demonstrating that responses in this model are enhanced and hypersynchronous [31] in addition to the impairment of auditory processing integration [32]. Electrical stimuli in the form of either cortical or DBS (Deep Brain Stimulation) have been used to monitor neuronal excitability and predict the occurrence of seizures in patients with epilepsy [7,25,33]. DBS for pre-ictal probing was also used in canines [34] and in a Pentilenotetrazole (PTZ) model in rats [35]. In the latter, probing effects were potentiated by previously pairing stimulation with ictal activity.

Seizures in isolated CA1 sections are preceded by the progressive increase of amplitude and duration of responses evoked by stimulation of Schaffer's collaterals [36]. This increased sensitivity and delayed recovery to perturbations characterizes the loss of a systems' stability and may be related to an approaching critical transition. It has been hypothesized that the onset of seizures would be ruled by this type of dynamics, sharing similarities with other types of complex phenomena and showing common early-warning signs such as critical slowing down, flickering, increased correlation, variability, and lag-1 autocorrelation (which is the estimate of the correlation of a signal with a 1-sample delayed version of itself) [37,38]. Contrasting evidence has been found both against [39–41] and in favor of this hypothesis [42–45]. Some questions regarding this might be elucidated with the use of perturbational approaches, which have been used to evaluate the existence and proximity to such tipping points in living dynamical systems [46] and could also be useful tools to explore the underlying dynamics of seizure onset in different models.

Neuronal computational models have been used extensively in the last decades, such as in trying to uncover the mechanisms responsible for brain rhythms [47], neuronal dynamics [48], coding [49], and pathologies like Epilepsy [50–54]. By defining sets of rules and

equations, models can be built to express either detailed or emerging general neuronal dynamics, offering insights about underlying mechanisms and enabling the testing of hypotheses that would be otherwise difficult or impossible to do experimentally. Individualized whole-brain models can be tailored to provide patient-specific interventions, such as in presurgical evaluation [55], or to predict and control seizure propagation [56,57]. Simulated microstimulation is proposed in [58] as a tool for tracking surround excitability levels to enable the operation of targeted intervention devices, as well as predicting surgical outcomes.

Established computational models provide a useful way of evaluating the effects of stimulation in neuronal systems [59,60]. Specifically, seizure-generating models permit exploration of how stimulation may be used for controlling seizures [61] or for the detection of transitions to ictal states with the aforementioned probing approach [9,62,63]. In both cases, the choice of stimulation parameters is a challenging task; widely different effects may be elicited by varying location, intensity, timing, frequency, polarity, and waveform of the stimulation component [59,64]. Thus, computational neuronal models allow for the evaluation of a wider range of stimulation parameters that would be cumbersome, if frankly infeasible, to do experimentally.

Despite the potential of this type of approach, few studies using probing to assess interictal to ictal transitions in neuronal computational models are found in the literature. Suffczynski et al. [62] used the model developed by Wendling et al. [65] to show that changes in excitability preceding the transition to a seizure are not observed in the ongoing EEG activity, but can be inferred by analysis combined with stimulation paradigms. Simulations also predicted the effects of electrical extracellular, local, bipolar stimulation (ELBS), showing that appropriately tuned stimulation activated preferentially GABAergic interneurons [59]. Furthermore, responses to ELBS in the computational model and patients with focal epilepsy were shown in this study to reveal hyperexcitable brain regions. The application of an active probing paradigm enabled a better observation of the underlying synchrony of a system of coupled Kuramoto oscillators [9], but this was only possible with suitably dense electrode recordings. That is, when electrodes are too spaced in-between, compared to the synchrony area, probing does not have the intended effect. The use of perturbation stimuli was also used with the Epileptor model to highlight feature changes signaling parameter changes towards ictal activity [63].

Using two coupled populations of Wendling's neural mass model, this work aims to explore different conditions in which an active probing approach is useful for highlighting the dynamical system changes occurring before seizure onsets. We expect that, as model parameters are shifted from normal towards ictal state, neuronal excitability and synchrony changes would be observable with the use of low frequency probing stimuli. Different model settings of coupled neural masses were evaluated, varying stimulation target and amplitude, as well as which model parameter is shifted to bring the system towards ictal states.

Results in this work corroborate previous literature evidence from computational and experimental data indicating that the impending risk of seizure states can be assessed using active probing approaches. Perturbation stimuli could highlight intrinsic excitability and/or synchrony changes that lead up to seizures, but this effect is limited to the neuronal population being stimulated. The observed feature changes such as increased variance and lag-1 autocorrelation are common hallmarks of the phenomenon of critical slowing down, which may be related to the dynamics underlying transitions to seizure states and can serve as early-warning signs of the imminence of such tipping points. And while stimuli did not elicit sustained discharges of spikes (thus indicating the lack of proictal effects), interictal-like discharges were elicited just before the transition to ictal activity under different model settings, indicating these could be robust markers for forecasting the imminence of seizures.

## Materials and methods

### CA1 neural mass model

Some neuronal computational models aim to describe in detail membrane potential and ion channel dynamics of specific neurons, such as the classic Hodgkin-Huxley model [66]. Networks of such detailed neurons interacting through synapses and gap junctions can be built, but their size and complexity are limited by both computational costs and dimension of their parameter space. Mesoscopic models may overcome some of these limitations by describing the mean activity of neuronal populations. Instead of individual variables for membrane potentials, action potentials (APs), or post-synaptic potentials (PSP) for each neuron, lumped descriptions of these attributes can be represented for specific subpopulations included in larger-scale networks [67], which are so far unfeasible to represent with detailed microscopic models. Pioneering works on such mean-field (or lumped-parameter) models include those from Wilson & Cowan [68], Freeman [69], Lopes da Silva et al. [70], and Wright & Liley [71]. Other models of this kind have been used to study spike-wave (SW) seizures [72] and their abatement using single-pulse stimulation [73].

This work is built upon the neural mass model developed by Wendling et al. [65], which consists of four coupled subpopulations and based on the global cellular organization of the CA1 subfield of the hippocampus; excitatory main cells (pyramidal cells), excitatory interneurons, inhibitory neurons with slow kinetics (O-LM neurons, with IPSCs mediated by dendritic synapses) and inhibitory neurons with fast kinetics (soma-projecting basket cells). In this work, an instance of this model is referred to as a neuronal population (or as a model subset), while specific neural masses (representing excitatory, slow inhibitory or fast inhibitory) are referred to as subpopulations. Interactions between these neuronal masses are represented in Fig 1, with excitatory interneurons being implicitly represented as the self-feedback loop from Pyramidal Cells. The average activity of each subpopulation is described by two main input-output relations, referred as "wave to pulse" and "pulse to wave" functions (i) The first function (wave to pulse) is an asymmetric sigmoid $S(v) = 2e_0/(1 + e^{r(v^0 - v)})$ which outputs the average efferent AP density of a population as a function of its average PSP or membrane potential. (ii) The second is a $2^{nd}$ order linear transfer function that relates the average PSP to the average density of incoming APs and can be either excitatory (EPSP) or inhibitory (IPSP). Impulse responses are represented by $h_e(t) = Aae^{-at}$ for excitatory synapses (EXC), $h_i(t) = Bbe^{-bt}$ for slow dendritic inhibition (SDI) and $h_g(t) = Gge^{-gt}$ for fast somatic inhibition (FSI).

The model receives an input signal $p_i(t)$, a normally distributed random variable and represents the density of afferent pulses or APs to the main excitatory (pyramidal) cells of neuronal population (or model subset) $i$. The model output is usually chosen as the average PSP of the main pyramidal cells, which presumably constitute the main generators of field potentials that make up the EEG signal.

### Model parameters and modifications

The main adjustable parameters of Wendling's neural mass model are usually A, B, and G. These refer to synaptic gains and influence the amplitude of the average excitatory (for A) or inhibitory (for B and G) PSPs. Changing these gives rise to at least six different activity types [65]; (1) normal background, (2) sporadic spikes, (3) sustained discharges of spikes, (4) slow rhythmic activity, (5) low-voltage fast activity and (6) slow quasi-sinusoidal activity.

In this work, two main modifications were made to the original model. The first is the bidirectional coupling of two neuronal populations (or model subsets) through excitatory main cells, as shown in Fig 1. This aims to represent inter-hemispheric hippocampal coupling,

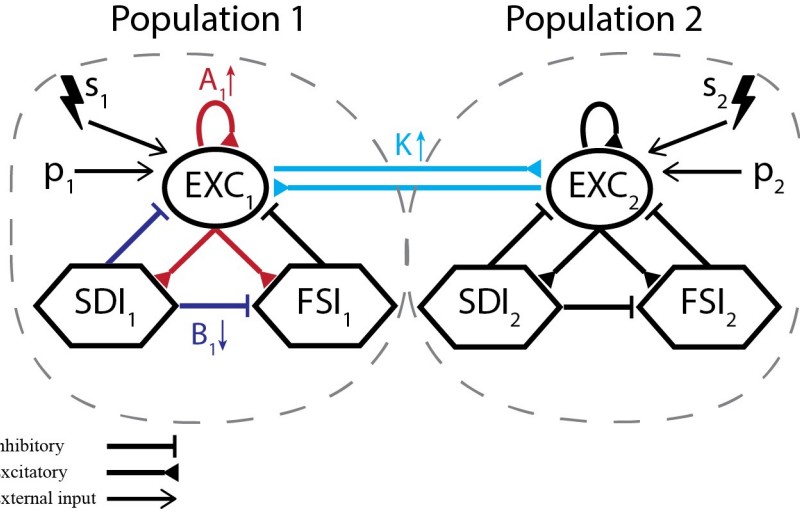

**Fig 1. Modified neural mass model.** (a) Representation of each coupled neuronal population and corresponding subpopulations. Excitatory interneurons are implicitly represented as the self-feedback loop from main pyramidal cells (EXC$_i$ to itself). Each model settings consists of either (I) unilateral ($s_2$) or (II) simultaneous bilateral low-frequency probing stimulation ($s_1$ and $s_2$), while slowly changing one parameter towards ictal activity–A$_1$, B$_1$ or K.

where each model subset or population represents the CA1 region of one cerebral hemisphere. This interaction is influenced by a coupling factor K (varied across simulations) and a delay coefficient $\tau_d$, set to 10 ms [74].

The second modification to the original model is the addition of low-frequency periodic stimuli. These probing stimuli are represented by $s_i(t)$ and summed to the model's Gaussian noise input $p_i(t)$, which is the incoming density of action potentials to the main excitatory cells of the neuronal population $i$. That is, the simplified modeling of the probing stimuli consists of an additional input function that represents incoming AP bursts due to the electrical stimulation of afferent regions such as the dentate gyrus or Schaffer collaterals. In the model, this is defined as periodic pulses with duration of 10 ms and amplitude $E_{stim}$, delivered every 2 seconds. This differs from the perturbation approach used previously by Suffczynski et al. [62], which simulated the placement of extracellular bipolar stimulation electrodes to change the membrane voltage of all subpopulations with pulses applied at a frequency of 10 Hz. We opted to use a slower stimulation frequency to ensure that the neuronal circuits are not entrained, preventing eventual pro-ictal effects due to the stimulus [75] (for a review on the pro or anti-convulsant effects of various stimulation parameters, see Ref. [76]).

The following set of 10 differential equations rules the activity of each neuronal population $i$.

$$\dot{y}_{i,0}(t) = y_{i,5}(t)$$

$$\dot{y}_{i,5}(t) = A_i a S[K_{i,j} y_{j,1}(t - \tau_d) + y_{i,1}(t) - y_{i,2}(t) - y_{i,3}(t)] - 2a y_{i,5}(t) - a^2 y_{i,0}(t)$$

$$\dot{y}_{i,1}(t) = y_{i,6}(t)$$

$$\dot{y}_{i,6}(t) = A_i a(s_i(t) + p_i(t) - C_2 S[C_1 y_{i,0}(t)]) - 2a y_{i,6}(t) - a^2 y_{i,1}(t)$$

$$\dot{y}_{i,2}(t) = y_{i,7}(t)$$

$$\dot{y}_{i,7}(t) = B_i b C_4 S[C_3 y_{i,0}(t)] - 2b y_{i,7}(t) - b^2 y_{i,2}(t)$$

$$\dot{y}_{i,3}(t) = y_{i,8}(t)$$

$$\dot{y}_{i,8}(t) = G_i g C_7 S[C_5 y_{i,0}(t) - C_6 y_{i,4}(t)] - 2g y_{i,8}(t) - g^2 y_{i,3}(t)$$

$$\dot{y}_{i,4}(t) = y_{i,9}(t)$$

$$\dot{y}_{i,9}(t) = B_i b S[C_3 y_{i,0}(t)] - 2b y_{i,9}(t) - b^2 y_{i,4}(t),$$

where $i \in [1,2]$ and refers to the number of the current neuronal population, $s_i$ is the probing stimulus applied to population $i$, and $K_{i,j}$ is the coupling factor between populations $i$ and $j$. In this work, bidirectional equal coupling was used, with $K_{1,2} = K_{2,1} = K$.

Values and descriptions of model parameters are given in Table 1.

In order to evaluate the effectiveness of probing stimuli to highlight preictal states, simulations are run with parameters set to normal activity, but linearly shifting one parameter towards the sustained discharges of spikes state throughout the simulation. This can be brought by simply increasing excitation (A) or decreasing inhibition (B) [65], but also by the time-varying interplay of interactions between excitatory and inhibitory synaptic gains [77]. In the modified model with coupled population subsets, the coupling factor K can also be used for seizure initiation. Thus, simulations were realized with default parameters as given in Table 1, but linearly varying $A_1$, $B_1$, or K in order to approach but not cross the threshold between normal and sustained spiking discharge activity.

The Euler-Maruyama method was used to solve the model's stochastic differential equations (SDE's), using a step size of 1.95 ms (sampling frequency of 512 Hz). The final time for each simulation is 2000 seconds. The model was implemented in Python 3.7, using the NumPy

**Table 1. Parameter descriptions and values [65].** A, B, and K values are either fixed or varied linearly throughout the simulations.

| Name | Description | Value |
|------|-------------|-------|
| $A_i$ | Average excitatory synaptic gain of population $i$ | 4.0 or [2.5 4.6] |
| $B_i$ | Average slow dendritic inhibitory gain of population $i$ | 40 or [45, 30] |
| $G_i$ | Average fast somatic inhibitory gain of population $i$ | 20 |
| K | Inter-population coupling factor | 0.3 or [0 0.5] |
| a | Dendritic average time constant in the feedback excitatory loop | 100 s$^{-1}$ |
| b | Dendritic average time constant in the slow feedback inhibitory loop | 50 s$^{-1}$ |
| g | Somatic average time constant in the fast feedback inhibitory loop | 350 s$^{-1}$ |
| C1, C2 | Mean number of synaptic contacts in the excitatory feedback loop | C1 = 135, C2 = 0.8 C1 |
| C3, C4 | Mean number of synaptic contacts in the slow feedback inhibitory loop | 0.25 C1 |
| C5, C6 | Mean number of synaptic contacts in the fast feedback inhibitory loop | 0.1 C1 |
| C7 | Mean number of synaptic contacts in between slow and fast inhibitory interneurons | 0.8 C1 |
| v0, e0, r | Parameters of the sigmoid (wave to pulse, or average PSP into average AP density) function | 6mV, 2.5 s$^{-1}$, 0.56 mV$^{-1}$ |
| μ, σ | Mean and standard deviation of noise input function | 90, 1.3 APs/s |
| $E_{stim}$ | Periodic probing stimuli amplitude, applied to the input of neuronal population $i$ | [0 200] APs/s |

[78], Joblib, scikit-learn [79], statsmodels, pandas [80], Matplotlib [81] and SciPy [82] packages. All scripts are available at https://github.com/vrcarva/WNMM_probing.

## Model settings

Different model settings were used to evaluate the effects of stimulation on predicting pre-ictal states, as illustrated in Fig 1. Each setting is characterized by the stimulation targets (one or both neuronal populations) and the parameter used to shift population activity towards ictal states–by increasing $A_1$ (EXC gain of population 1), decreasing $B_1$ (SDI gain of population 1), or increasing K (inter-population coupling gain). Detailed parameters of each setting are given next. All other parameters are set according to Table 1.

i. **Probing neuronal population 2**

 A. Increasing $A_1$: $A_1$ = 2.5 to 4.6, $A_2$ = 4.0, B = 40, K = 0.3

 B. Decreasing $B_1$: A = 4.0, $B_1$ = 45 to 30, $B_2$ = 40, K = 0.3

 K. Increasing K: A = 4.0, B = 40, K = 0 to 0.5

ii. **Probing neuronal populations 1 and 2**

 A. Increasing $A_1$: $A_1$ = 2.5 to 4.6, $A_2$ = 4.0, B = 40, K = 0.3

 B. Decreasing $B_1$: A = 4.0, $B_1$ = 45 to 30, $B_2$ = 40, K = 0.3

 K. Increasing K: A = 4.0, B = 40, K = 0 to 0.5

Thus, model setting I consists of probing population 2 while set 1 parameters are shifted towards ictal-like activity. This aims to evaluate if applying stimuli to different regions that are not ictogenic (but coupled to ictogenic ones) allows the identification of impending ictal transitions. Unilateral probing of the ictogenic population 1 is evaluated in additional model settings, included in S1 and S2 Figs. The use of bilateral probing is evaluated in model setting II, where both populations receive stimuli simultaneously. Fig 1 illustrates the interactions between neuronal populations and subpopulations of the proposed model settings.

## Evaluating probing effectiveness

After each simulation, different features were extracted from 400 ms post-stimuli epochs of each neuronal population's output. The set of features is composed of variance, skewness, kurtosis and lag-1 autocorrelation (lag-1 AC) of each simulated LFP, and mutual information between the activity of both coupled populations to assess synchronization [83,84]. A moving average filter of order 20 was used for smoothing the resulting feature time series.

In each simulation, the dynamics of the extracted features may indicate if the system is approaching the transition to an ictal state. In addition to visual inspection, where increasing or decreasing feature trends could indicate increased seizure susceptibility, it is important to quantify how good predictor a feature is and compare this quantitative measure with and without probing stimulation. This can be done by assessing the relationship between the shifted parameter and the selected feature. The Spearman's rank correlation coefficient [85] was used for this in order to account for non-linear correlations. It is defined by:

$$\rho = \frac{\sum_k^n (x_k - \bar{x})(y_k - \bar{y})}{\sqrt{\sum_k^n (x_k - \bar{x})^2}\sqrt{\sum_k^n (y_k - \bar{y})^2}},$$

where $k$ is the paired score, $x$ and $y$ are the ranks of extracted features (variance, skewness, kurtosis, Lag-1 AC, and mutual information) and shifted parameters ($A_1$, $B_1$, or K), respectively.

To evaluate the sensitivity of probing efficiency as a function of stimulus intensity, input pulse amplitude is gradually increased from 0 (i.e., passive paradigm) to 200 APs/s, in steps of 20. In each test, fifteen realizations were run, yielding distributions for each extracted feature time-series, from which correlations measures were calculated. This enables statistical comparison between the effects of increasing probing stimuli amplitude–this was done with Tukey's honestly significant difference (HSD) method, accounting for pairwise comparisons of probing efficiency between increasing stimulation amplitudes and the passive observation case (stimulus amplitude = 0).

A crucial assessment that must be done is whether the stimuli have pro-ictal effects. That is, probing stimuli should help to predict impending seizures, not elicit them. Thus, additional simulations were done with the same model settings and varying degrees of stimulus amplitude, but shifting parameters to elicit ictal activity in the form of sustained discharges of spikes. For each simulation, seizure onset was defined as the first time in the simulation where spike discharges occurred in succession (more specifically, five spikes with an inter-spike interval of at least 4 seconds occur in succession). Spikes were detected if a peak was found in the simulated signal with amplitude greater than 5 mV. In order to exclude stimuli responses, peaks found up to 200 ms after perturbation onset were discarded. Once again, multiple comparisons testing with Tukey's HSD were made to test if stimulation of increasing amplitudes resulted in significantly different seizure onset times in relation to the passive case with no stimuli.

## Results

Simulated signals are shown in Fig 2, with and without stimulation of both neuronal populations (120 APs/sec). First, the parameter $A_1$ was varied linearly from 4.3 to 4.95, eliciting ictal activity at the end of the simulation. As a seizure approaches, the shape of the evoked responses changes gradually, up to a point in which the response is drastically increased. These evoked responses can be considered as interictal like spikes (ILS) [86] evoked by the stimuli when the system is approaching an ictal-like state. They are akin to afterdischarges encountered during electrical stimulation. There was no evidence that a sustained series of spontaneous discharges (e.g. an ictal run) was generated. Only unitary events were observed.

### Resulting features and probing effectiveness

Probing was then evaluated with model setting I, in which a population is made ictogenic (but without crossing the threshold to ictal activity) by shifting a parameter linearly throughout each simulation–increasing $A_1$ (I-A), decreasing $B_1$ (I-B), or increasing K (I-K)–while perturbation stimuli are applied to the input of population 2. Fig 3 shows the resulting feature series for I-A for simulations without probing, followed by the Spearman's rank correlation measure between $A_1$ and each extracted feature series for various degrees of stimulus intensity.

For model setting I-A, the approaching transition to ictal activity of the ictogenic population 1 can be observed even without probing, due to the increase in signal variance. However, increasing stimulus amplitude (greater than 140 APs/s) affects other features for the "normal" population 2 –slight decreases in skewness and kurtosis can be observed with the approaching ictal transition, only when probing stimuli are used, suggesting a predictive effect of the perturbational approach in this setting. High stimulation amplitudes also highlight the increased mutual information between both model subsets.

For model setting I-B, Fig 4 shows slight changes in Lag-1 AC and variance (and to a lesser extent, in kurtosis as well) of population 1, regardless of stimulation. That is, as the SDI gain is decreased, no predictive effects of active probing are observed in the activity of the neuronal populations, and neither in the mutual information between both simulated signals.

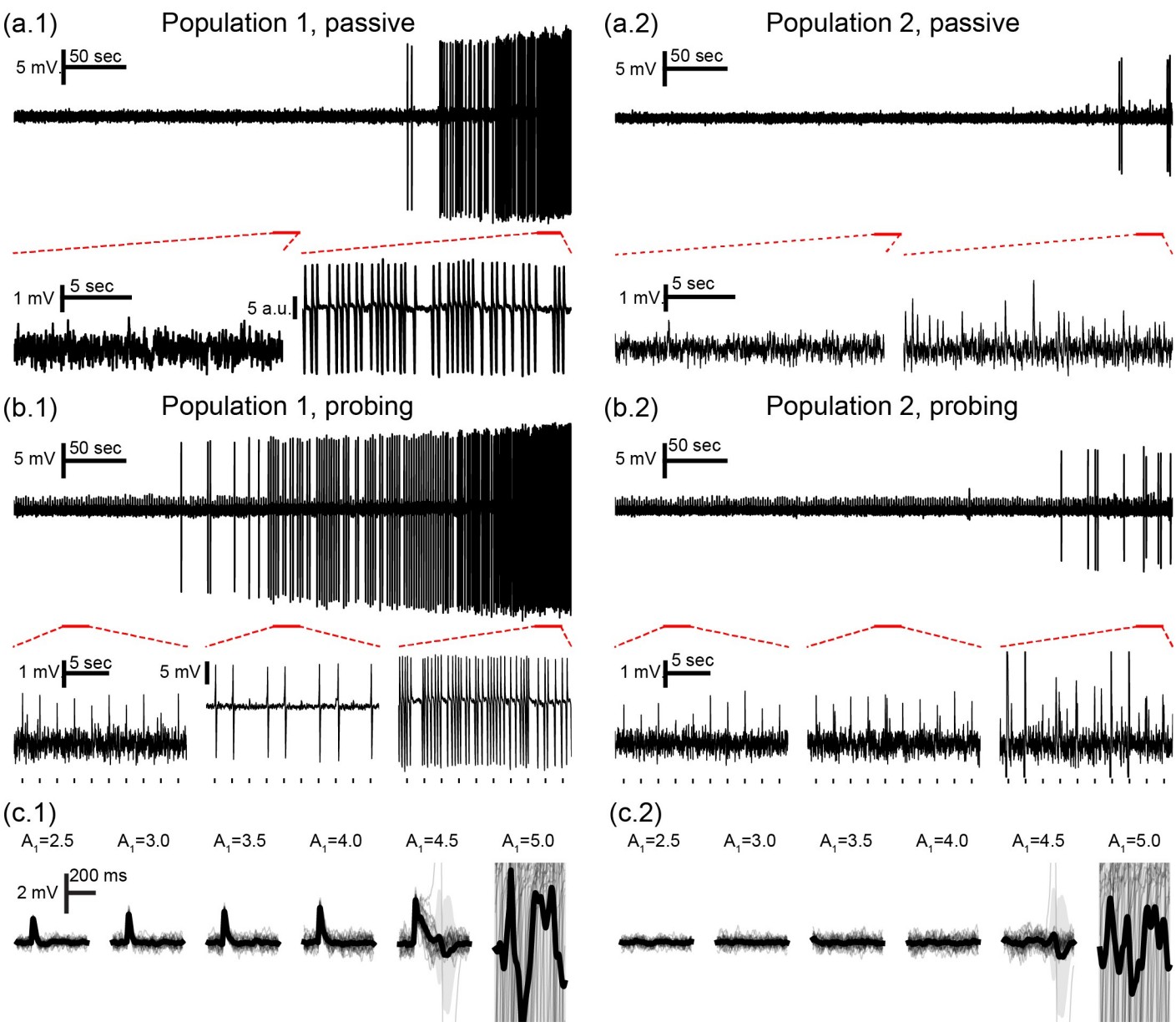

**Fig 2. Neuronal mass model outputs with and without active probing stimuli.** Simulation changing parameter $A_1$ from 4.30 to 4.95 (B = 40, G = 20) to elicit ictal activity in the end, with (a) passive observation or (b) active probing of both populations. Outputs are shown for population 1 (a.1 and b.1) and population 2 (a.2 and b.2), with tick marks showing stimuli onsets for (b). Ictal-like activity in the form of sustained discharges of spikes is observed for population 1 at the end of both simulations. ILS appear before this transition and are elicited even further in advance by stimulation. (c) Bottom plots show overlapped and mean response waveforms for increasing values of $A_1$ for (c.1) population 1 and (c.2) population 2. Gradual response changes are observed until the appearance of ILS for $A_1 = 4.5$ and ictal activity at $A_1 = 5.0$.

For I-K, all features from the stimulated population correlate with the increased coupling factor K–and consequently with the time to ictal activity onset. However, as shown in Fig 5, these changes are not observed with the un-perturbed population 1. Altogether, the results for setting I indicate that the effectiveness of active probing to highlight preictal changes is limited to the model subset being perturbed. However, if unilateral stimulation is applied to the icto-genic population, as shown in S1 and S2 Figs, increased feature changes can be detected in both populations.

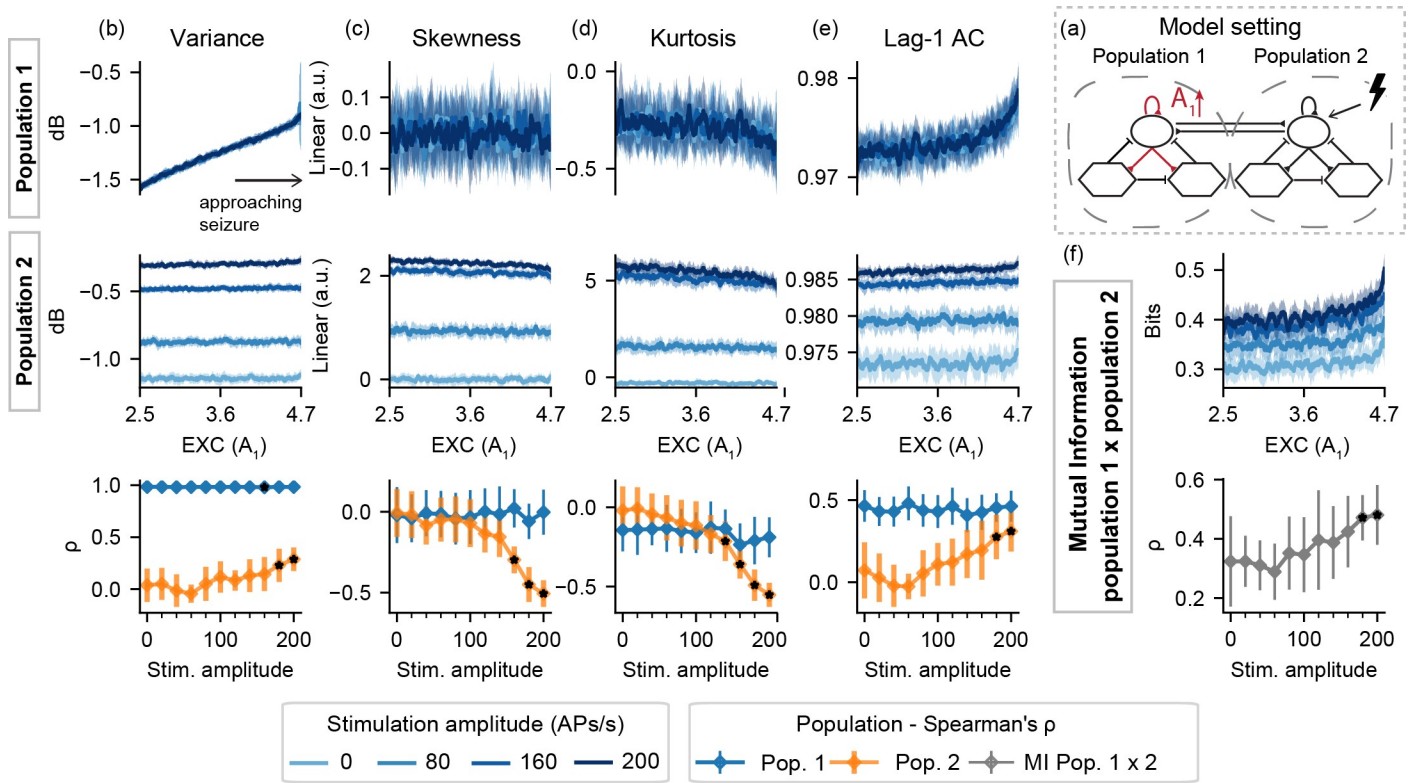

**Fig 3. Feature series and probing efficiency for model setting I-A.** For population 1, excitability increase is correlated with variance even with passive observation. However, increased amplitude probing responses from population 2 reflect gradual changes of $A_1$ towards ictogenesis. (a) shows a simplified illustration of the model setting. For (b) variance, (c) skewness, (d) kurtosis and (e) lag-1 AC features, top rows show feature values extracted from the output of each population, as the excitability gain parameter ($A_1$) is increased. (f) shows the inter-population synchronization as measured by the mutual information feature. Bottom row shows Spearman's rank correlation between each feature series and the shifted parameter ($A_1$), as stimulus amplitude is increased. Black dots indicate significant differences from passive observation (stim. Amplitude = 0) according to Tukey's HSD.

Next, results are shown for model settings II, where both populations are simultaneously probed while one parameter is shifted towards ictal activity. This is done with the increase of $A_1$ (EXC), the decrease of $B_1$ (SDI), or the increase of coupling both populations (K). Fig 6 shows the resulting feature series and Spearman's rank correlation for model setting II-A.

For higher stimulation amplitudes in model setting II-A, feature changes are evident as the transition to ictal activity approaches. This can be seen in both populations, albeit to a lesser degree in the "normal" model subset, which parameters are held constant. Higher stimulus amplitudes result in sudden increases in variance near the threshold from normal to ictal activity, reflecting the appearance of increased sharp responses in the form of ILS as shown in Fig 2. Furthermore, increased stimuli amplitudes highlight increased mutual information between the simulated activities of both populations.

For the ictogenic population 1 in setting II-B, Fig 7 shows that active probing induces significant feature changes as the transition to ictal state approaches. Once again, the sharp increase in signal variance for higher stimulation amplitudes reflects enhanced responses or ILS shown in Fig 2. Preictal changes are much less visible in the "normal" simulated LFP from population 2, appearing only for skewness and kurtosis with stimulation amplitudes greater than 140 APs/s. On the other hand, consistent increases in mutual information between the activities of both populations are revealed by bilateral stimuli greater than 80 APs/s.

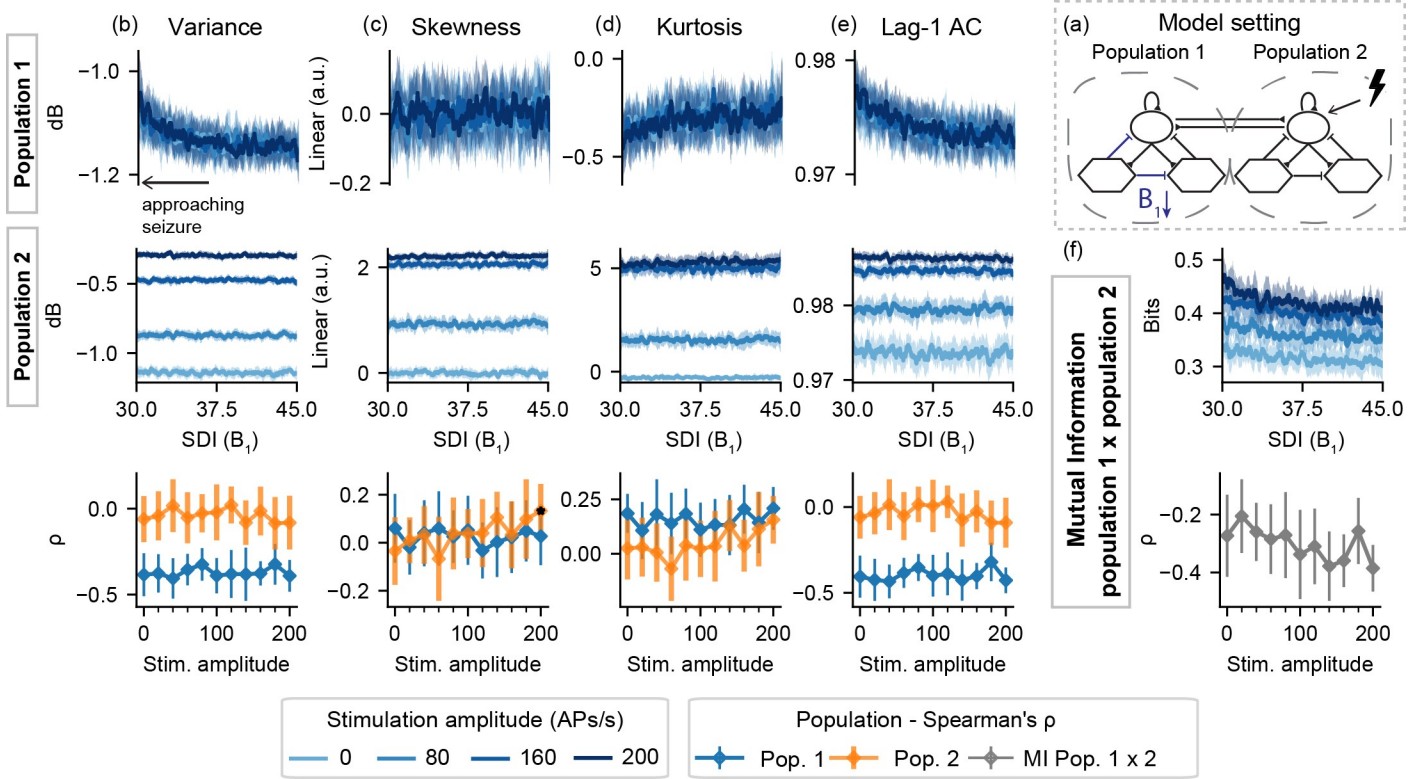

**Fig 4. Feature series and probing efficiency for model setting I-B.** Decreased slow inhibition of population 1 is not highlighted by the extracted features if only the "normal" neural mass (set 2) is stimulated. (a) shows a simplified illustration of the model setting. For (b) variance, (c) skewness, (d) kurtosis and (e) lag-1 AC features, top rows show feature values extracted from the output of each population, as the slow inhibition gain parameter ($B_1$) is decreased. (f) shows the inter-population synchronization as measured by the mutual information. Bottom row shows Spearman's rank correlation between each feature series and the shifted parameter ($B_1$), as stimulus amplitude is increased.

Similar feature profiles can be observed in Fig 8 with model setting II-K–sharp increases in variance, lag-1 AC, and mutual information and decrease in skewness and kurtosis, but only when probing stimuli are used.

## Testing for pro-ictal effects of active probing

To ensure that the probing stimuli have no pro-ictal effects, additional simulations were made to evaluate if the occurrence of ictal-like activity is precipitated by the probing stimuli. The same model settings were used, but parameters were shifted further to elicit sustained spike discharges–Fig 9 shows the onset times of this state for each population as a function of stimulus amplitude.

One-way ANOVA tests yielded overall significant differences in ictal-like activity onset times for model settings II-A ($p = 0.028$) and II-B ($p = 0.000231$). However, Tukey's HSD of all stimulation amplitudes versus the passive observation case (amplitude = 0) did not result in any significant differences. That is, stimulation did not anticipate the occurrence of sustained discharges of spikes when compared to the passive observation case, implying that the probing approach did not have pro-ictal effects in any of the evaluated model settings and stimulation parameters.

## Discussion

Using different model settings of coupled populations of a neural mass model, we have explored the ability of active probing to highlight early warning signs that reflect intrinsic

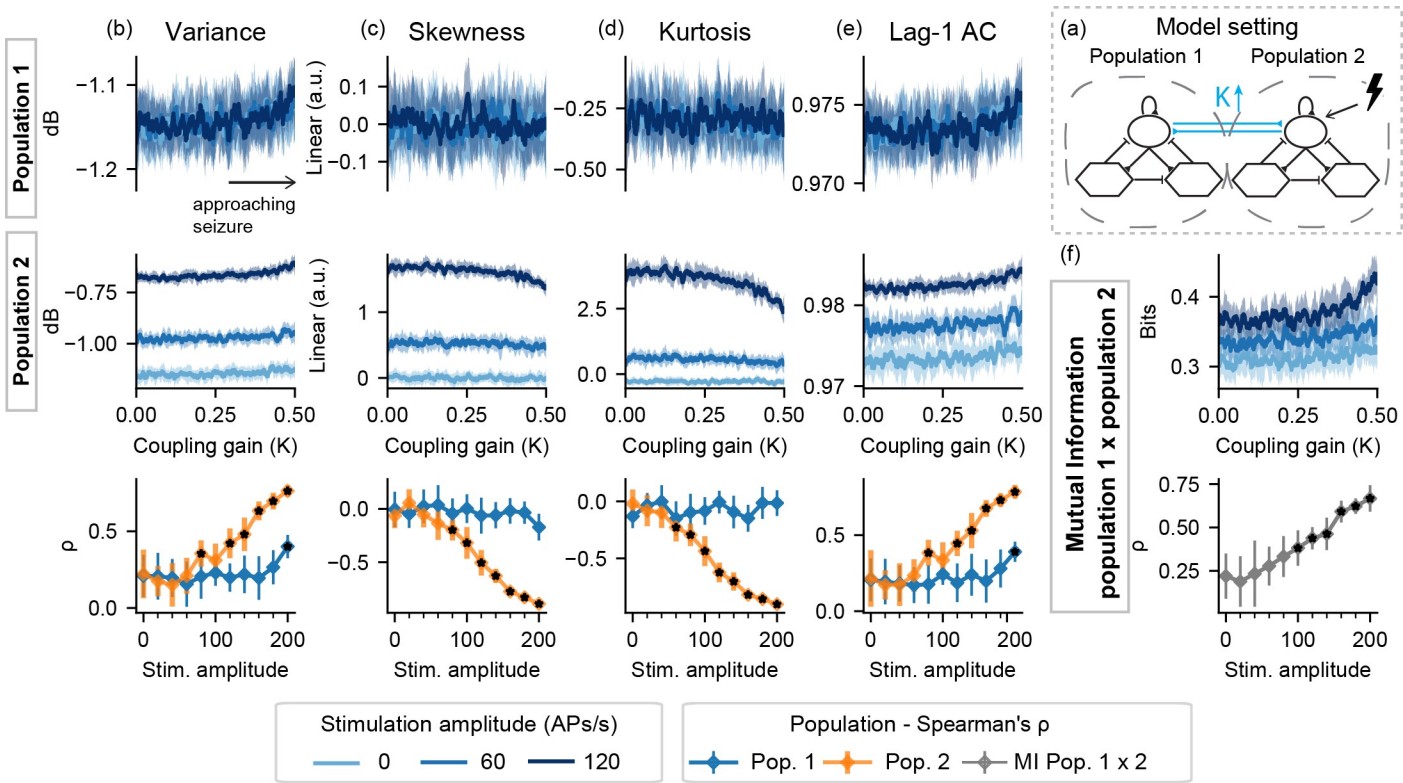

**Fig 5. Feature series and probing efficiency for model setting I-K.** Coupling gain increase is highlighted by features extracted from the stimulated population and by the mutual information between activities of both model subsets. (a) shows a simplified illustration of the model setting. For (b) variance, (c) skewness, (d) kurtosis and (e) lag-1 AC features, top rows show feature values extracted from the output of each population, as the coupling gain gain parameter (K) is increased. (f) shows the inter-population synchronization as measured by the mutual information feature. Bottom row shows Spearman's rank correlation between each feature series and the shifted parameter (K), as stimulus amplitude is increased.

parameter changes involved in triggering ictal states. In most model settings, the perturbational approach induced response changes that could be used as markers of changing parameters that lead to ictal activity–mainly the increase in variance and lag-1 autocorrelation and decreases in skewness and kurtosis of the simulated signals, as well as the increase in the inter-population synchrony. Importantly, a local effect of active probing was observed, as these changes are not observed in features extracted from the un-perturbed populations, except when unilateral stimuli are applied to the ictogenic population.

In model setting I, extracted features (namely variance and lag-1 AC) from the ictogenic population 1 are slightly sensitive to the decrease of slow inhibition ($B_1$) and very sensitive to the increase in excitability ($A_1$), even with passive observation. For features extracted from the probed population 2, the effects depended on which parameter was being altered. No detectable changes were found in setting I-B, but increases in the coupling factor (K) and excitability ($A_1$) were correlated with features extracted from the perturbed population 2. In the case of I-A, the effects were subtle and restricted to high stimulation amplitudes. That is, the effects of active probing were local, being limited to the population being stimulated. This reaffirms the findings in Ref. [9], which highlighted the importance of suitably dense recordings for the active probing approach to be successful in extracting system information.

In model setting II, the predictive effects of probing the ictogenic population are observed regardless of which parameter shift was used to bring the model dynamics towards ictal activity–extracted features from stimuli responses are correlated with the increasing excitability

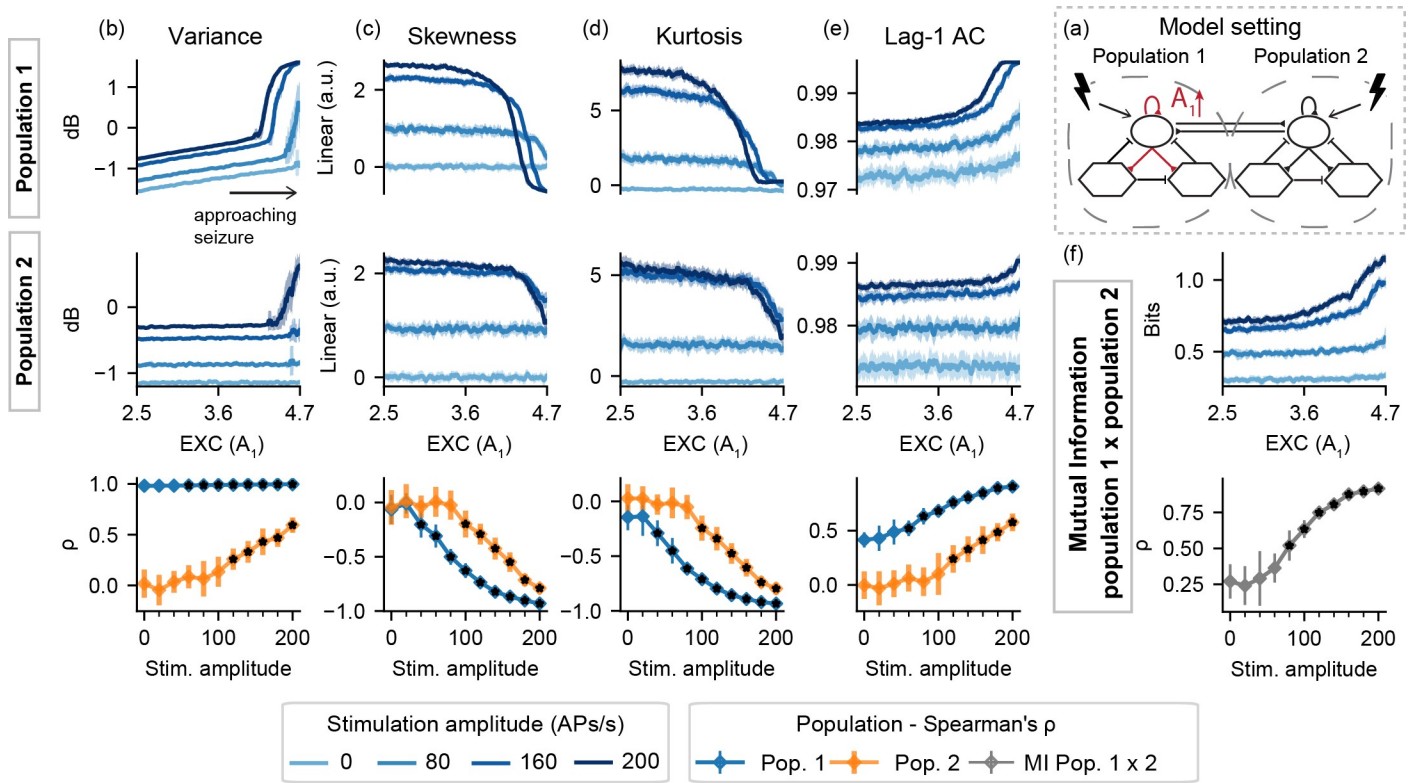

**Fig 6. Feature series and probing efficiency for model setting II-A.** Active probing effects are visible for population 2, but only for increased stimulation amplitudes. (a) shows a simplified illustration of the model setting. For (b) variance, (c) skewness, (d) kurtosis and (e) lag-1 AC features, top rows show feature values extracted from the output of each population, as the excitability gain parameter ($A_1$) is increased. (f) shows the inter-population synchronization as measured by the mutual information feature. Bottom row shows Spearman's rank correlation between each feature series and the shifted parameter ($A_1$), as stimulus amplitude is increased.

($A_1$) or decreasing slow inhibition ($B_1$), as well as the increase in coupling strength ($K$). Except for the increase in variance as $A_1$ is increased, these changes are not visible with the passive observation of population activity. That is, the results indicate that the use of active probing to reveal approaching transitions from normal to ictal activity is very effective when stimuli are applied to the ictogenic population. Furthermore, the perturbational approach may benefit from the use of coordinated stimulation in different regions, since changes heralding the approach of ictal activity were visible not only in the ictogenic population activity, but also in the coupled neuronal populations with fixed parameters (as in the case of model setting II-A). This is especially relevant, considering that refractory epilepsy is increasingly being considered as a network disease [87]. The combination of stimulation of the ictal onset zone and remote structures has been shown previously to further increase DBS efficacy in epilepsy patients [88]. Network topology also plays a role in seizure generation and may be used for planning optimal surgery strategies–e.g., scale-free and rich-club networks display specific nodes that are critical for seizure generation and should be good candidates for removal [89]. Similar studies could show how different active probing configurations and parameters should be tailored for specific network topologies in order to leverage predictive power while assuring no pro-ictal effects occur.

In this work, external perturbation of a neural mass model highlighted common signatures of approaching critical transitions such as increased synchronization, variance, and lag-1 AC [38]. The increase in probing response variance and lag-1 autocorrelation is similar to the

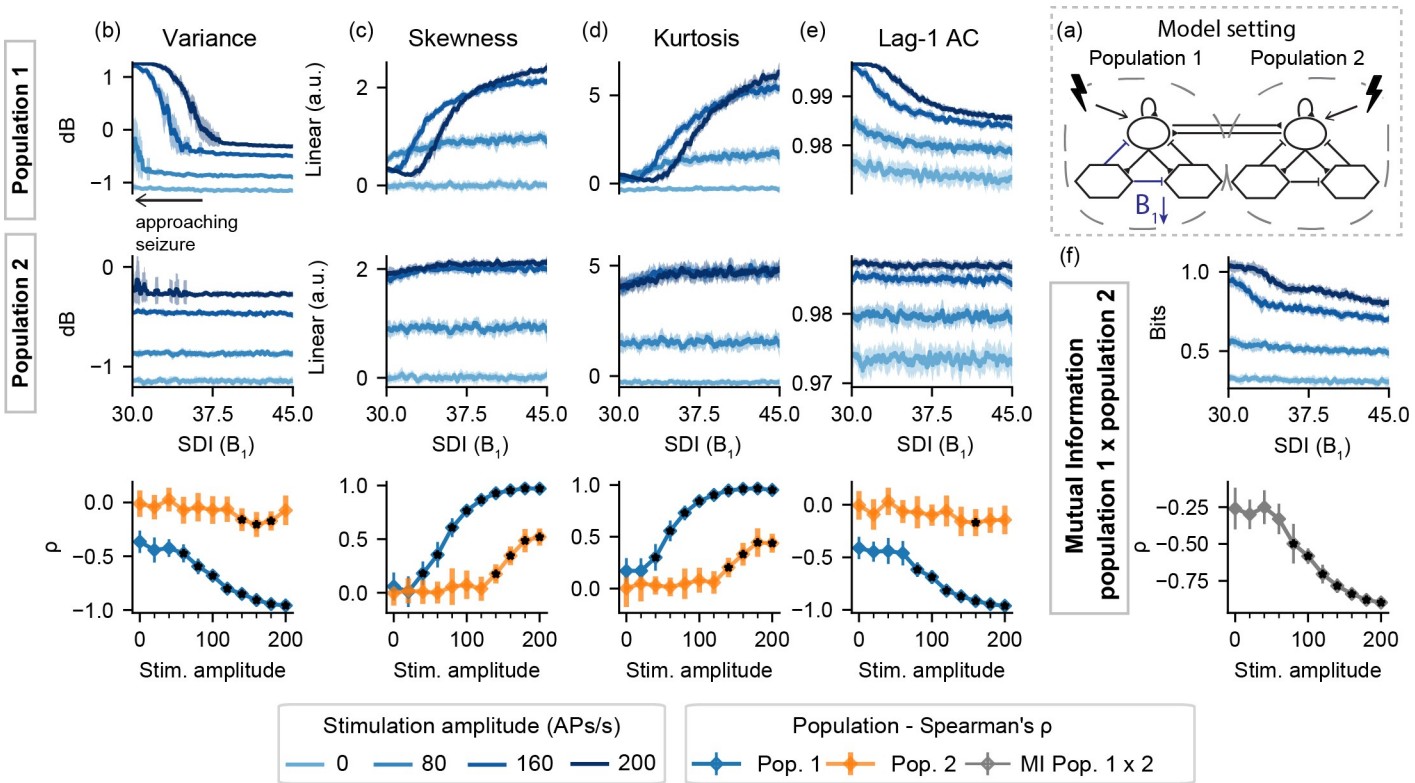

**Fig 7. Feature series and probing efficiency for model setting II-B.** Active probing provides predictive value for features extracted from both populations, but the effect is limited to increased stimulation amplitudes for population 2. (a) shows a simplified illustration of the model setting. For (b) variance, (c) skewness, (d) kurtosis and (e) lag-1 AC features, top rows show feature values extracted from the output of each population, as the slow inhibitory gain parameter ($B_1$) is decreased. (f) shows the inter-population synchronization as measured by the mutual information feature. Bottom row shows Spearman's rank correlation between each feature series and the shifted parameter ($B_1$), as stimulus amplitude is increased.

results found in Chang *et al.* [36], which examined the stability of the system's response to stimulation of the Schaffer collaterals in isolated CA1 sections. The increased response amplitudes and longer recovery times from the perturbations were interpreted as markers of decreasing system resilience and critical slowing down. This is a common early warning signal of an impending critical transition, which are sudden shifts in dynamic states that have been reported in a myriad of complex dynamical systems [90] and are marked by common feature patterns [37]. It is still a matter of debate whether seizures would belong to this class of phenomena. Although it has been shown that human seizure termination can occur via critical transition [91], conflicting evidence has been found if this is true for the initiation of seizures. Comparing a model exhibiting self-organized criticality (SOC) with experimental data, deviations of a power-law were observed during human epileptic seizures and could indicate a shift of dynamics toward an ordered phase during ictal events [40]. Several other computational models involve critical transitions leading to seizure onsets [7,43,44,92] and experimental evidence in favor of this is found in Refs [36,45]. However, only a minority of patient seizures have been shown by Milanowski & Suffczynski [39] to present classical hallmarks or critical transitions before seizure onset. In another study, long-lasting iEEG recordings from 28 epilepsy patients did not reveal evidence for critical slowing down prior to epileptic seizures in humans [41]. On the other hand, a significant increase of lag1-AC was observed before seizure onset in 4 of 12 patients in Ref. [36], suggesting that critical slowing down should be evident in periods leading up to seizures in at least specific populations of epilepsy patients. This is also

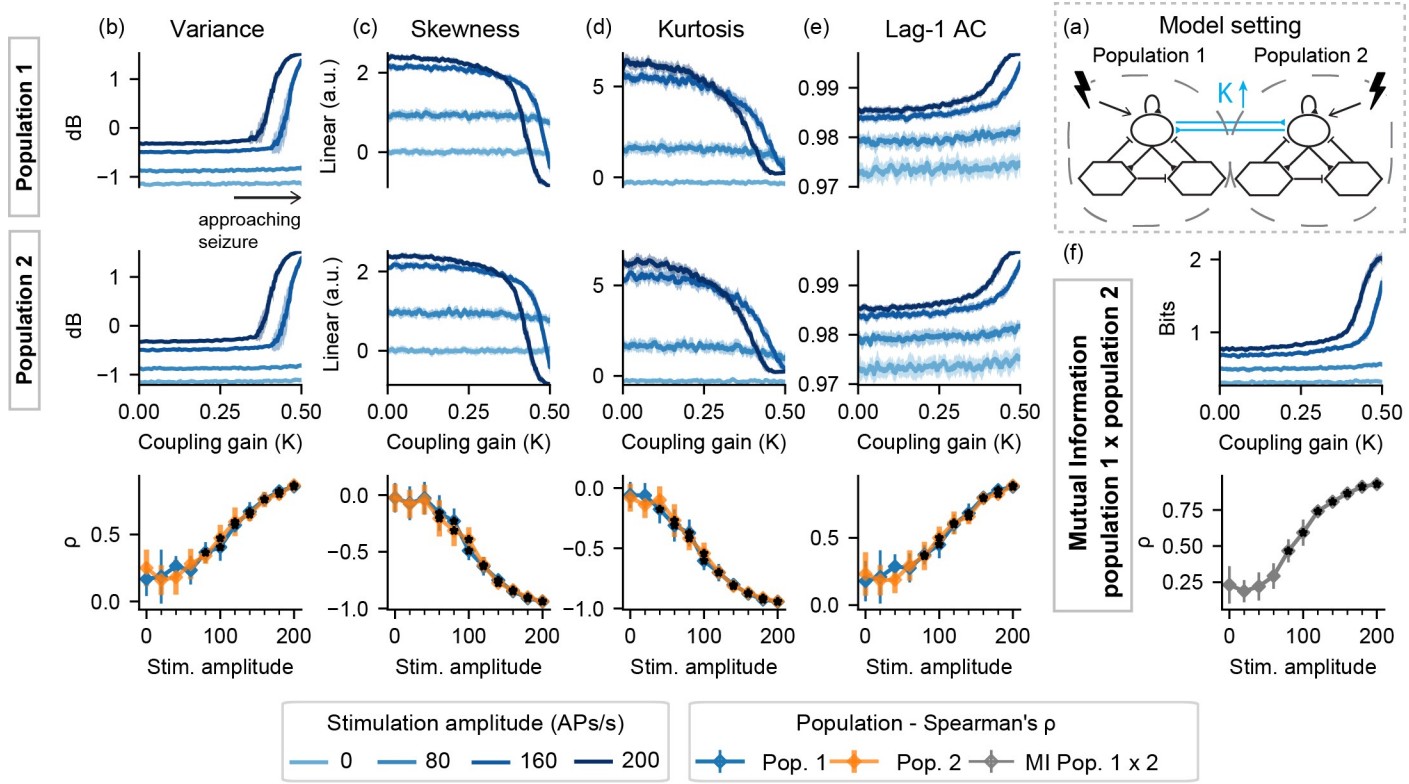

**Fig 8. Feature series and probing efficiency for model setting II-K.** Probing provides predictive value for both stimulated populations. (a) shows a simplified illustration of the model setting. For (b) variance, (c) skewness, (d) kurtosis and (e) lag-1 AC features, top rows show feature values extracted from the output of each population, as the coupling gain parameter (K) is increased. (f) shows the inter-population synchronization as measured by the mutual information feature. Bottom row shows Spearman's rank correlation between each feature series and the shifted parameter (K), as stimulus amplitude is increased.

supported by a recent work which found that warning signals of critical slowing down (namely variance and autocorrelation) changed before seizure onset in 13 of 14 epilepsy patients over small temporal scales, but observed changes were also found in longer temporal scales (hours and days, as opposed to commonly observed scales of seconds to minutes) for 9 of 14 patients [42].

Critical slowing down can be evidenced by increased variance and longer recovery times from perturbations such as periodic stimuli or even by noise inputs [43]. This makes the use of probing stimuli an interesting tool to evaluate if seizure onsets are characterized by critical transitions in different computational and experimental seizure models, which would make its associated classical hallmarks promising markers for seizure forecasting, especially when combined with other features such as the rate of epileptiform spikes [42].

Active probing approaches may be viable but not needed in some cases where increased seizure susceptibility can be measured by intrinsic electrographic features, such as the increased signal variance of a population that has increased excitability parameter ($A_1$), considering the model evaluated in this work. Intrinsic measures were used by Meisel et al. [93] to show how cortical excitability is reduced by antiepileptic drugs and increased across the wake period in patients. These measures were correlated with stimulation-evoked responses, supporting their viability as markers of cortical excitability and could play a role in seizure forecasting algorithms.

In model settings where stimuli response features were correlated with the shifted parameters, features evolved either gradually (as in I-K), or gradually until drastic changes appeared

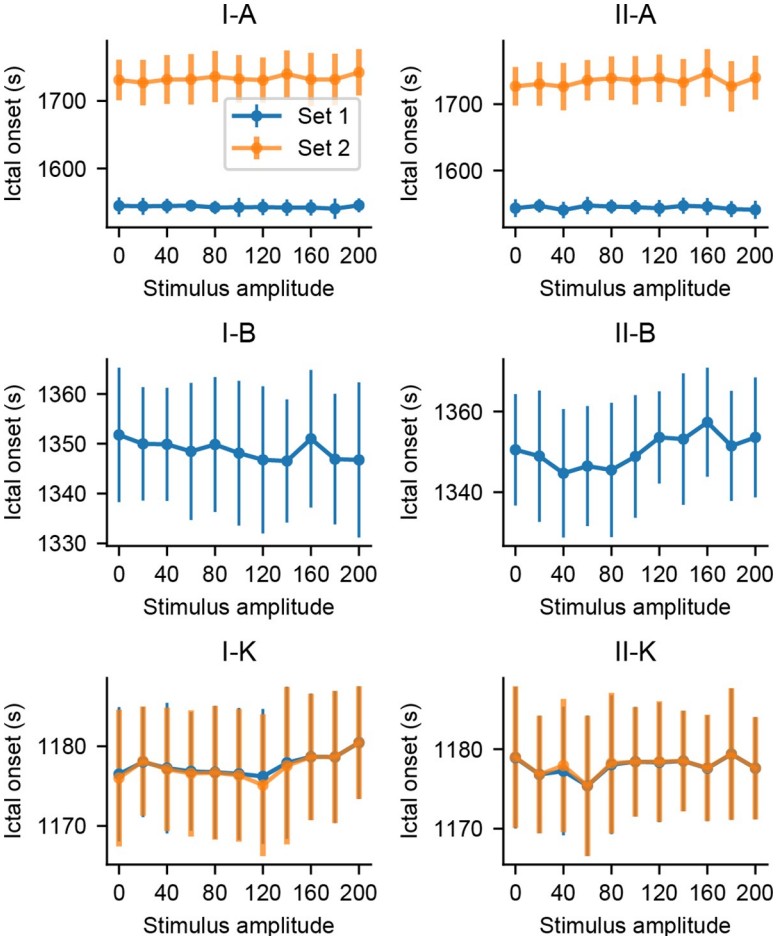

**Fig 9. Onset time of ictal-like activity as a function of stimulus amplitude.** Simulations with increased amplitudes did not affect the onset latencies of sustained spike discharges in any model setting.

before ictal onset (II-K for high stimuli intensities). The latter changes are due to the appearance of sharp stimuli responses near the threshold to ictal activity and could be interpreted as interictal-like spikes (ILS) elicited by stimulation (see Fig 2). The results here are similar to the findings by Lopes *et al.* [86], which studied the dynamics of interictal spikes and early-warning signs of approaching critical transitions in a stochastic cellular automata neural network model (SCANNM). It was shown that the required external stimulation to evoke ILS became smaller as the model's activity approached the ictal state, such that optimal stimuli intensity values could thus be tuned to elicit ILS in the minimally sufficient time for serving as clear early-warning signals of impending seizures. Electrical stimuli also appeared to induce ILS right before seizure onset in a PTZ seizure rat model in Medeiros *et al.* [45].

Probing stimuli were found to have no pro-ictal effect in the neural mass model used in this work. That is, sustained discharges of spikes were not precipitated by the use of low-frequency period stimulation. However, responses similar to ILS were evoked with increased stimuli intensities before the transition to ictal activity. While these responses can serve as clear early-warning signs, the role of interictal discharges in epilepsy and whether these events are anti or pro-ictal is still unclear [94–96]. Thus, despite the lack of pro-ictal effects of the active probing approach in this work and others [25,45,59], further experimental evidence is needed to ensure

that the use of long-term probing stimuli would not induce pro-ictal effects while still providing seizure-forecasting value.

## Conclusion

The lack of clinically validated and robust methods for seizure forecasting motivates the evaluation of active probing approaches, where the use of stimuli may be used to highlight electrographic features indicating impending ictal state transitions. By using a neural mass model with two coupled populations and different parameter changes leading up to seizures, this work showed that a low-frequency active probing approach provides predictive value. As parameters of a neuronal population were shifted from normal towards ictal activity, responses showed changes that were in general not visible with the passive observation of the populations' activity, such as increased variance, lag-1 autocorrelation and synchrony between neuronal populations' activities. A local effect of active probing was observed, if stimulation is applied to the 'normal' neuronal population. That is, activity dynamics heralding approaching transitions to ictal states are limited to the features being extracted from the stimulated population. The feature changes were consistent with the phenomenon of critical slowing down, where response variance and lag-1 autocorrelation increased as the model approached the ictal state. This is often related to the loss of a systems' resilience and are common hallmarks of an impending critical transition. Furthermore, increasing stimulation amplitude did not cause significant changes in the onset times of sustained discharges of spikes, indicating the lack of pro-ictal effects of active probing in this neural mass model. Altogether, results indicate that low-frequency probing stimuli can provide predictive value and reveal underlying dynamics involved in transitions from normal to ictal states.

The optimistic projections that in the next few years part of the burden of epilepsy in some patients may be eased by seizure forecasting methods are motivated by current evidence of the feasibility of anticipating seizures and recent advances in the field [12]. New methods may leverage insights gained not only from active probing approaches discussed here, but also from increasingly available long-term iEEG databases [97], as well as increased adoption of new implantable devices that enable patient-tailored algorithms. Achieving this would be a crucial step that may enable to ultimately close the loop for targeted seizure-controlling interventions.

## Supporting information

**S1 Fig. Feature series and probing efficiency for unilateral probing of ictogenic population with increased excitability.** (a) shows a simplified illustration of the model setting. For (b) variance, (c) skewness, (d) kurtosis and (e) lag-1 AC features, top rows show feature values extracted from the output of each population, as the excitability gain parameter ($A_1$) is increased. (f) shows the inter-population synchronization as measured by the mutual information feature. Bottom row shows Spearman's rank correlation between each feature series and the shifted parameter ($A_1$), as stimulus amplitude is increased. Black dots indicate significant differences from passive observation (stim. Amplitude = 0) according to Tukey's HSD. For population 1, excitability increase ($A_1$) is correlated with variance even with passive observation. Increased correlation of other features, including mutual information between population activities, with the ictogenic parameter $A_1$ occurs only with stimulation. For features extracted from population 2, high stimulation amplitudes result in changes prior to ictal onset, even though this population was not stimulated–probably due to the ILS elicited in population 1.
(TIF)

**S2 Fig. Feature series and probing efficiency for unilateral probing of ictogenic population with decreased slow inhibition.** (a) shows a simplified illustration of the model setting. For (b) variance, (c) skewness, (d) kurtosis and (e) lag-1 AC features, top rows show feature values extracted from the output of each population, as the excitability gain parameter ($A_1$) is increased. (f) shows the inter-population synchronization as measured by the mutual information feature. Bottom row shows Spearman's rank correlation between each feature series and the shifted parameter ($B_1$), as stimulus amplitude is increased. Black dots indicate significant differences from passive observation (stim. Amplitude = 0) according to Tukey's HSD. For population 1, SDI decrease ($B_1$) is slightly correlated with variance even with passive observation, but stimulation increases the correlation with this and other features, including mutual information between population activities. For features extracted from population 2, high stimulation amplitudes result in changes prior to ictal onset, even though this population was not stimulated–probably due to the ILS elicited in population 1.
(TIF)

## Acknowledgments

We would like to thank Daniel Medeiros, Flávio Mourão and Leonardo Guarnieri for helpful comments and discussions on the manuscript.

## Author Contributions

**Conceptualization:** Vinícius Rezende Carvalho, Márcio Flávio Dutra Moraes, Eduardo Mazoni Andrade Marçal Mendes.

**Formal analysis:** Vinícius Rezende Carvalho.

**Funding acquisition:** Márcio Flávio Dutra Moraes, Eduardo Mazoni Andrade Marçal Mendes.

**Investigation:** Vinícius Rezende Carvalho.

**Methodology:** Vinícius Rezende Carvalho.

**Project administration:** Márcio Flávio Dutra Moraes, Eduardo Mazoni Andrade Marçal Mendes.

**Resources:** Márcio Flávio Dutra Moraes, Sydney S. Cash, Eduardo Mazoni Andrade Marçal Mendes.

**Software:** Vinícius Rezende Carvalho.

**Supervision:** Márcio Flávio Dutra Moraes, Sydney S. Cash, Eduardo Mazoni Andrade Marçal Mendes.

**Validation:** Vinícius Rezende Carvalho, Sydney S. Cash, Eduardo Mazoni Andrade Marçal Mendes.

**Visualization:** Vinícius Rezende Carvalho.

**Writing – original draft:** Vinícius Rezende Carvalho.

**Writing – review & editing:** Vinícius Rezende Carvalho, Márcio Flávio Dutra Moraes, Sydney S. Cash, Eduardo Mazoni Andrade Marçal Mendes.

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
