## [Decision Letter · Decision Letter 0]

5 Nov 2020

Dear Prof. Mendes,

Thank you very much for submitting your manuscript "Active probing to highlight approaching transitions to ictal states in coupled neural mass models" for consideration at PLOS Computational Biology.

As with all papers reviewed by the journal, your manuscript was reviewed by members of the editorial board and by several independent reviewers. In light of the reviews (below this email), we would like to invite the resubmission of a significantly-revised version that takes into account the reviewers' comments.

We cannot make any decision about publication until we have seen the revised manuscript and your response to the reviewers' comments. Your revised manuscript is also likely to be sent to reviewers for further evaluation.

Sincerely,

Peter Neal Taylor

Associate Editor

PLOS Computational Biology

Daniele Marinazzo

Deputy Editor

PLOS Computational Biology

Reviewer's Responses to Questions

**Comments to the Authors:**

Reviewer #1: The article discusses the important issue of predicting epileptic discharges using the existing computational neural mass model. A feature of this work is the analysis of signals received from the model. The task of predicting discharges is solved by modeling external stimuli by changing the parameters of the model.

The disadvantages of the work include:

0. In the "abstract" section, authors should better identify the methods that were used in this particular work.

1. Using the term lag-1 autocorrelation without clarification at the initial mention.

2. Not all known mesoscale EEG models are listed on line 124 (for example https://doi.org/10.1371/journal.pone.0114316, https://doi.org/10.1007/BF00202790, https://doi.org/10.1371/journal.pone.0239125).

3. On line 229, it is not entirely clear what is "a moving average filter of size 20".

4. Line 253. Please clarify what does "5 au" mean.

5. In my opinion, the "Conclusions" section should be written in a little more detail.

It should be emphasized that new was obtained by analyzing the signals of the existing Wendling model.

In general, the work is done and described at a good level, the results are presented clearly and in sufficient detail.

Reviewer #2: The Ms describes application of neural mass model to study stimulation-based assessment of system closeness to seizure. The stimulation-based biomarkers of approaching transitions are based on a critical slowing down signal measures and are shown be able to identify proximity of the ictal state. The study is original and offers useful, conceptual insight into the problem of seizure prediction using active paradigms. The authors are familiar with the literature relevant to the problem and describe their work in this context. In general it is very good work that deserves a publication. I have only minor comments to improve the Ms .

Minor points:

1. The authors use 2 model settings – with unilateral and bilateral stimulation. The unilateral stimulation is delivered only to the side with the ‘normal’ set of parameters. It seems counterintuitive. The most direct test would be to selectively stimulate the population with ‘pathological’ set of parameters to see if the probing stimulus can identify the on-going process of ictogenesis. Such test is not performed. Please explain your rationale of probing ‘normal’ network rather than ‘epileptic’ one in the unilateral setting.

2. Figure 4, top row, shows that Variance, Lag-1 AC and to some extend Kurtosis measured in Population 1 are sensitive to B parameter change (I-B setting). In the text it is described that feature changes indicating change of B were not found except AC(1). Please clarify.

3. The meaning of black dots in Spearman rank correlations in all figures is not explained.

4. Figures 4, 5, 7, 8 are not referred in the text.

5. Model settings are named Ia, Ib etc (ln. 205-212). They are referred to as I-A, I-B later in the text. Please unify.

6. Figure 2 is a bit confusing, partly because of panel annotations a.1, a.2, b.1 etc. I suggest to mark the panels with headings e.g. ‘Population 1’, ‘Population 2’, ‘Active’, ‘Passive’ etc. It will help to quickly get information from the figure without going through the (complex) legend. Also the role of enlarged signals is not clear. What is the reader supposed to learn from them? I suggest to simplify the figure by removing them or at least making them smaller.

7. Setting III-A is mentioned in ln. 378. It is not described in the Model setting section.

**Have all data underlying the figures and results presented in the manuscript been provided?**

Reviewer #1: Yes

Reviewer #2: Yes

PLOS authors have the option to publish the peer review history of their article (what does this mean?). If published, this will include your full peer review and any attached files.

Reviewer #1: No

Reviewer #2: No
---

## [Decision Letter · Decision Letter 1]

2 Dec 2020

Dear Prof. Mendes,

We are pleased to inform you that your manuscript 'Active probing to highlight approaching transitions to ictal states in coupled neural mass models' has been provisionally accepted for publication in PLOS Computational Biology.

Best regards,

Peter Neal Taylor

Associate Editor

PLOS Computational Biology

Daniele Marinazzo

Deputy Editor

PLOS Computational Biology

Reviewer's Responses to Questions

**Comments to the Authors:**

Reviewer #1: All the points that I raised were adequately addressed.

Reviewer #2: None

**Have all data underlying the figures and results presented in the manuscript been provided?**

Reviewer #1: Yes

Reviewer #2: Yes

PLOS authors have the option to publish the peer review history of their article (what does this mean?). If published, this will include your full peer review and any attached files.

Reviewer #1: No

Reviewer #2: No

---

## [Editor Report · Acceptance letter]

19 Jan 2021

PCOMPBIOL-D-20-01787R1 

Active probing to highlight approaching transitions to ictal states in coupled neural mass models

Dear Dr Mendes,

I am pleased to inform you that your manuscript has been formally accepted for publication in PLOS Computational Biology. Your manuscript is now with our production department and you will be notified of the publication date in due course.

With kind regards,

Jutka Oroszlan
